# The Impact of Different Antihypertensive Drugs on Cardiovascular Risk in Isolated Systolic Hypertension with Type 2 Diabetes Patients

**DOI:** 10.3390/jcm11216486

**Published:** 2022-11-01

**Authors:** Ming Gao, Wenrui Lin, Tianqi Ma, Yi Luo, Hejian Xie, Xunjie Cheng, Yongping Bai

**Affiliations:** 1Department of Geriatric Medicine, Center of Coronary Circulation, Xiangya Hospital, Central South University, Changsha 410008, China; 2Department of Dermatology, Xiangya Hospital, Central South University, Changsha 410008, China; 3National Clinical Research Center for Geriatric Disorders, Xiangya Hospital, Central South University, Changsha 410008, China

**Keywords:** diabetes mellitus, isolated systolic hypertension, angiotensin receptor blockers, cardiovascular mortality

## Abstract

Backgrounds: Angiotensin receptor blockers (ARB), angiotensin converting enzyme inhibitor (ACEI), calcium channel blocker (CCB) and thiazide diuretics (TD) are common antihypertensive drugs for diabetes patients with hypertension. The purpose of this study was to compare the cardiovascular risks of these drugs in patients with isolated systolic hypertension (ISH) and type 2 diabetes mellitus (T2DM). Methods: We used Action to Control Cardiovascular Risk in Diabetes trial data to explore the relationship between antihypertensive drugs and cardiovascular risks in ISH with T2DM patients by performing propensity score matching, Kaplan–Meier survival analyses and Cox proportional regression. Results: The cumulative incidence rates of primary outcomes (PO, including cardiovascular mortality, non-fatal myocardial infarction and non-fatal stroke) in the ARB use group were significantly lower than those without (hazard ratio (HR) 0.53; 95% confidence interval (CI) 0.34–0.83; *p* = 0.006). However, for ACEI, CCB and TD, they were negligible (ACEI: *p* = 0.209; CCB: *p* = 0.245; TD: *p* = 0.438). ARB decreased cardiovascular mortality (CM) in PO rather than non-fatal myocardial infarction (NMI) and non-fatal stroke (NST) (CM: HR 0.32; 95%CI 0.18–0.90; *p* = 0.004; NMI: *p* = 0.692; NST: *p* = 0.933). Conclusion: ARB may alleviate the cardiovascular risks in ISH with T2DM patients, but ACEI, CCB, and TD did not.

## 1. Introduction

Type 2 diabetes mellitus (T2DM) with isolated systolic hypertension (ISH) is a type of diabetes mellitus with high cardiovascular risk characterized by increased pulse pressure (PP). The high cardiovascular risk of people with diabetes is reflected in the fact that diabetes can cause atherosclerosis, leading to coronary artery disease, cerebrovascular disease, peripheral artery disease and other cardiovascular diseases [1,2,3]. The cause of elevated pulse pressure in T2DM patients is related to increased vascular stiffness caused by impaired carbohydrate metabolism [4]. This may be the reason why people with type 2 diabetes are more likely to develop ISH. The prevalence of ISH in the general population ranges from 1.9% to 4.3% [5,6]. In a cross-sectional study of Chinese patients with type 2 diabetes, the prevalence of ISH was found to be 7.6%, more than twofold higher than that of non-diabetic patients (3.4%), and the incidence and mortality of cardiovascular diseases are much higher than the general population [7,8,9,10]. A decreasing of systolic blood pressure can significantly improve the prognosis of diabetes, and a decrease of 10 mmHg leads to 12% reductions in diabetes complications (myocardial infarction, sudden death, angina, stroke, renal failure, lower extremity amputation or death from peripheral vascular disease, death from hyperglycemia or hypoglycemia, heart failure, vitreous hemorrhage, retinal photocoagulation, and cataract extraction) [11,12]. Therefore, it is important for improving prognosis to control blood pressure in T2DM patients.

Drug therapy is a common way to control blood pressure in diabetes with hypertension patients, and there are a variety of antihypertensive drugs available. Angiotensin receptor blockers (ARBs) and angiotensin-converting enzyme inhibitors (ACEIs) are recommended for reducing proteinuria and slowing renal decline in T2DM patients [13]. Thiazide diuretics (TDs) and calcium channel blockers (CCBs) are effective in reducing the risks of stroke and of other morbid events in hypertension patients [14]. All of these drugs are often recommended for the antihypertensive treatment of diabetes with hypertension patients. However, we must realize that the problem is that T2DM with ISH patients have greater pulse pressure, a higher degree of arterial blood wall stiffness, a more serious degree of vascular aging, and greater cardiovascular risk [9]. Therefore, it is not clear whether these drugs are also suitable for T2DM with ISH patients, and there is a lack of research on the effect of using and not using these antihypertensive drugs on their cardiovascular risk. We decided to use the data from the ACCORD trial for a post-hoc analysis to investigate the effects of four antihypertensive drugs (ARB, ACEI, CCB or TD) on the cardiovascular prognosis of ISH with T2DM patients, so as to provide a scientific basis for clinical rational drug use.

## 2. Materials and Methods

### 2.1. Study Design

To evaluate the effects of first-line antihypertensive drugs in T2DM [15] on cardiovascular risk in ISH with T2DM, we retrospectively analyzed the data from ACCORD, which was funded by the National Heart, Lung, and Blood Institute. The detailed rationale, design and results of the ACCORD trial have been published previously [16].

The ACCORD trial, a prospective multi-center randomized controlled double two-by-two factorial clinical trial conducted in America and Canada, recruited 10,251 middle-aged and older volunteers diagnosed with T2DM for at least 3 months according to the 1997 American Diabetes Association (ADA) criteria. The ACCORD main trial recruitment started in February 2003. The final visit for the last randomized participant was ended on 30 June 2009, with each participant receiving 4 to 8 years of treatment and follow-up (approximate mean, 5.6 years). All participants were randomly assigned to groups of standard glycemia control (HbA1c target of 7–7.9%) or intensive glycemia control (HbA1c target of ≤6.0%). Hypoglycemic medical therapy mainly includes sulfonylureas, biguanides, meglitinide, alpha-glucosidase inhibitors, thiazolidinediones, regular insulins and so on. The primary outcome measure for ACCORD is the first occurrence of a major cardiovascular event (nonfatal myocardial infarction, nonfatal stroke, or cardiovascular death). Secondary outcomes include other cardiovascular outcomes, total mortality, health-related quality of life, and cost-effectiveness. We have added a detailed CONSORT-flow diagram in Appendix A. According to European and American guidelines, isolated systolic hypertension was defined as systolic blood pressure ≥ 140 mmHg and diastolic blood pressure < 90 mmHg [15,17]. The number of people who met the criteria of isolated systolic hypertension was 2547, and the prevalence was about 24.84%. The ACCORD dataset is available from the National Heart, Lung, and Blood Institute Biologic Specimen and Data Repository (https://biolincc.nhlbi.nih.gov/studies/accord/). The use of the ACCORD dataset in this study has been approved by the National Heart, Lung, and Blood Institute and the institutional review board of Xiangya Hospital, Central South University.

### 2.2. Outcome Measurements

In this study, outcomes of interest included primary outcomes (POs), cardiovascular mortality (CM), non-fatal myocardial infarction (NMI) and non-fatal stroke (NST). CM was defined as fatal stroke, myocardial infarction, heart failure, arrhythmia, sudden cardiovascular death within 24 h, and death from other vascular diseases such as pulmonary embolism [16]. POs included the first occurrence of CM, NMI or NST. All results were reviewed by two reviewers without knowledge of investigator assignments and associated treatment strategies.

### 2.3. Statistical Analysis

According to the baseline status of antihypertensive drugs that patients used and did not use, we separated ISH with T2DM individuals into four large groups: use and non-use ARB group, use and non-use ACEI group, use and non-use CCB group and use and non-use TD group. Then, we divided each large group into two small groups: the treatment group (use ARB group, use ACEI group, use CCB group and use TD group) and the non-treatment group (non-use ARB group, non-use ACEI group, non-use CCB group and non-use TD group). We adjusted the confounding and potential selection bias between the treatment group and non-treatment group by propensity score and the 1:1 nearest neighbor matching method [18]. To reduce the effects of increased loss to follow-up when the follow-up time was above seven years, the cumulative events rate of outcomes arising within seven-year visits were analyzed. Variables matched between these two groups are listed in Table 1. The continuous variables in the study are consistent with the normal distribution, and Student’s t test was used for the continuous variables. Categorical variables have been compared by Chi-square analysis.

Kaplan–Meier survival analyses and log-rank test were used to evaluate the cumulative survival rate of primary outcomes. The independent correlations between different antihypertensive drugs and primary outcomes have been evaluated by Cox proportional regression analysis, shown as hazard ratio (HR) with 95% confidence interval (CI). The initial analysis was unadjusted (Model 1). The adjusted HR and 95% CI were computed in Model 2 by adding the baseline age and gender to Model 1, and in Model 3 by adding race, education, Body Mass Index (BMI), smoking history, systolic and dilated blood pressure, HbA1c level, lipid level (low-density lipoprotein, high-density lipoprotein and triglyceride), therapies, previous heart failure, previous CVD, sulfonylureas, biguanides, meglitinide, alpha-glucosidase inhibitors, thiazolidinediones, regular insulins and statins to Model 2. Cox proportional regression analysis was also used for the subsequent independent correlation evaluations between ARB and CM, and NMI and NST. A two-tailed probability value less than 0.05 was considered statistically significant. All data used in this study were analyzed in R 4.1.1.

## 3. Results

### 3.1. Clinical Characteristics of Patients

The baseline characteristics of patients are shown in Table 1. In total, 1062 patients were in the use and non-use ARB group, and the ages of participants with or without ARB were 64.83 ± 6.95 and 64.97 ± 6.74, respectively. A total of 2230 ISH with T2DM patients were enrolled in the use and non-use ACEI group, and the ages of the use ACEI group and non-use ACEI group were 64.55 ± 6.74 and 64.58 ± 6.63. There were 1266 patients in the use and non-use CCB group, and the ages were 65.76 ± 6.96 and 65.38 ± 6.64. The number in the use and non-use TD group was 1612, and the ages of the use TD group and non-use TD group were 64.86 ± 6.79 and 64.72 ± 6.43, respectively. After propensity score and 1:1 nearest neighbor matching, the variables that significantly yielded selection bias and misunderstanding results were well matched, including age, gender, BMI, HbA1c level, smoking history, lipids, blood pressure, race, education, previous heart failure, previous CVD, sulfonylureas, biguanides, meglitinide, alpha-glucosidase inhibitors, thiazolidinediones, regular insulins and statins. Other factors that could potentially influence the subjects’ outcomes were also well matched.

### 3.2. Cardiovascular Outcomes and Mortalities

Figure 1 showed Kaplan−Meier survival analysis of primary outcomes (POs) in T2DM with ISH patients treated with and without antihypertensive drugs. POs were lower in subjects treated with ARB than those without ARB therapy (*p* = 0.046), which suggests an association between ARB use and decreased PO. However, for ACEI, CCB and TD, there were no differences between patients with antihypertensive therapy intervention and those without (ACEI: *p* = 0.11; CCB: *p* = 0.27; TD: *p* = 0.51).

As shown in Table 2, Cox proportional regression analysis was used to further evaluate the risk of outcomes. Consistent with the results of the Kaplan–Meier survival analysis, ARB treatment in Model 1, Model 2, and Model 3 reduced the risk of PO in T2DM with ISH patients (Model 3: HR 0.53; 95%CI 0.34–0.83; *p* = 0.006). The impacts that other antihypertensive therapies exerted on POs, however, were negligible (ACEI: *p* = 0.209; CCB: *p* = 0.245; TD: *p* = 0.438). In Appendix A, we have added the effects of four antihypertensive drugs on the total mortality of T2DM with ISH patients, but unfortunately, they did not improve the total mortality (Model 3: ARB: *p* = 0.125; ACEI: *p* = 0.137; CCB: *p* = 0.759; TD: *p* = 0.876).

Since “primary outcomes” was a composite outcome, we separately evaluated the relationship between ARB therapy and CM, NMI and NST (Table 3). CM was significantly lower in patients with ARB therapy than those without in Model 1, Model 2, and Model 3 (Model 3: HR 0.32; 95%*CI* 0.18–0.90; *p* = 0.004). However, the cumulative event rate of NMI and NST in individuals receiving the ARB therapy was similar to that for those without therapy (NMI: *p* = 0.692; NST: *p* = 0.933).

## 4. Discussion

Our study retrospectively analyzed the data from the ACCORD trial to reveal the relationship between different antihypertensive medications and cardiovascular outcomes in T2DM with ISH patients. In this study, we identified that ARB treatment was associated with a decrease in PO in T2DM with ISH patients. However, there was no significant difference in POs between patients treated with ACEI, CCB, and TD, and those not treated. In addition, our further analysis showed that ARB decreased CM rather than NMI and NST in the compound outcome of PO. These results indicate that ARB was superior to other common antihypertensive drugs in the selection of antihypertensive drugs for T2DM with ISH patients, and can significantly reduce the risk of cardiovascular events and mortality.

ARB is one of the recommended antihypertensive drugs for patients with diabetes mellitus complicated with hypertension. A meta-analysis of randomized controlled studies showed that treatment with ARB results in a significant reduction in cardiovascular events and mortality in hypertensive patients with T2DM [19]. This was similar to the conclusion of our study, which found that ARB can reduce the incidence of cardiovascular events and improve the prognosis of T2DM with ISH patients (HR 0.53; 95%CI 0.34–0.83; *p* = 0.006). On the one hand, this may be related to the fact that ARB can target peroxisome proliferator-activated receptor (PPAR-γ) to improve insulin sensitivity and glucose metabolism in diabetic patients [20,21]. On the other hand, this may be related to the renal protective effect of ARB, in addition to its antihypertensive effect. ARB can reduce the expression of heme oxygenase, inhibit the formation of pentoglycoside to reduce proteinuria, prevent glomerular and tubulointerstitial injury, protect the renal function of diabetic patients and improve the prognosis [22]. However, another randomized double-blind placebo-controlled trial showed that ARBs did not reduce cardiovascular events in diabetic patients with hypertension [23]. When they later analyzed the effects of baseline covariates on outcomes, they found that this was probably related to having more men in the placebo group.

ACEIs have similar mechanisms of action to ARB; both can inhibit the renin angiotensin aldosterone system (RAAS) to play an antihypertensive role, and are also recommended as antihypertensive drugs for diabetic patients [24]. However, the effects of these two drugs on cardiovascular risk in diabetic patients with hypertension have been controversial. Some studies found that ACEI can reduce the mortality of patients with diabetes, while ARB cannot. ACEI should remain the preferred inhibitor of RAAS in high-risk diabetes mellitus and hypertension patients [25]. However, a real-world study in Taiwan showed that ARB may be more suitable than ACEI for first-line antihypertensive drug selection in T2DM patients. The incidence of stroke was 26% lower in the ARB group than in the ACEI group, and ARB was more suitable for primary stroke prevention [26]. This may be related to the fact that ARB can improve insulin sensitivity in diabetic patients, but ACEI cannot [27]. Our findings also show that ARB can effectively reduce the incidence of cardiovascular events and cardiovascular mortality in T2DM with ISH patients, while ACEI did not (HR 1.19; 95%CI 0.91–1.54; *p* = 0.209).

CCB can also reduce blood pressure variability in T2DM patients with hypertension in addition to lowering blood pressure, so it is recommended as an antihypertensive agent [28,29]. However, the effect of CCB on cardiovascular risk in diabetic patients with hypertension remains controversial. A meta-analysis by Ettehad et al., of 123 studies and 613,815 participants, confirmed the superiority of CCBs in stroke prevention in T2DM with hypertension [30]. This is inconsistent with our view. Our study shows that CCB did not reduce cardiovascular events or mortality in T2DM with ISH patients (HR 1.23; 95%CI 0.87–1.73; *p* = 0.245). Another meta-analysis of randomized trials has also reported that CCB treatment did not improve mortality in diabetic patients with hypertension [31]. The possible explanation is that although CCB can reduce blood pressure and reduce renal injury, on the other hand, it can enhance sympathetic nerve activity and cause vascular dilation of afferent renal arterioles, promoting proteinuria and increasing renal injury [32]. The two may be offset to varying degrees, leading to the lack of advantages of CCB in reducing cardiovascular risk in diabetic patients.

TD is one of the best choices of antihypertensive drugs in patients with diabetes mellitus complicated with hypertension [33,34,35]. Some studies have revealed that TD could reduce the incidence of major cardiovascular events in diabetic patients, especially those with obesity or fluid retention [36]. However, an online meta-analysis of 27 randomized controlled trials (RCTS) with 49,418 participants showed no benefit of TD therapy in reducing all-cause or cardiovascular mortality in T2DM with hypertension patients [31]. This was similar to our findings that TD use did not improve the incidence of cardiovascular events in T2DM with ISH patients (HR 0.88; 95%CI 0.64–1.21; *p* = 0.438). The cardiovascular effects of TD on diabetic patients are double-sided. For one thing, TD can reduce fluid retention and relieve the load on the heart. For another, TD can cause metabolic disorders (for example, it causes electrolyte disturbances, leading to hypokalemia, hyponatremia, and hypomagnesemia, affects fasting glucose, and leads to a deterioration of glucose tolerance, exacerbating hepatic steatosis, with increased liver triglyceride content, leading to visceral fat accumulation and decreased insulin sensitivity) and interfere with the cardiovascular prognosis of diabetes patients [37,38,39,40]. This may be the reason why the effect of TD on the cardiovascular prognosis of diabetic patients has manifested differently in different studies.

There were some limitations in this study. First, potential bias has been reduced by using the propensity score and 1:1 nearest neighbor matching. However, the propensity score method was only applicable to known confounding factors, and could not reduce the effects of unknown cardiovascular factors. Secondly, this study did not concern the combination of medication in ISH with T2DM patients, and this is the direction we need to pay attention to in further studies. Finally, the ACCORD trial data did not reveal information about the doses and timings of ARB, ACEI, CCB and TD, which may lead to bias and necessitate further prospective randomized clinical trials to verify our conclusions.

## 5. Conclusions

Among ISH with T2DM patients, the cumulative incidence rate of POs in the ARB use group was significantly lower than in those in the no ARB use group. Compared with ACEI, CCB and TD, antihypertensive therapy with ARB may be an effective method to improve clinical prognosis and reduce cardiovascular mortality. However, further clinical studies are needed to confirm this.

## Figures and Tables

**Figure 1 jcm-11-06486-f001:**
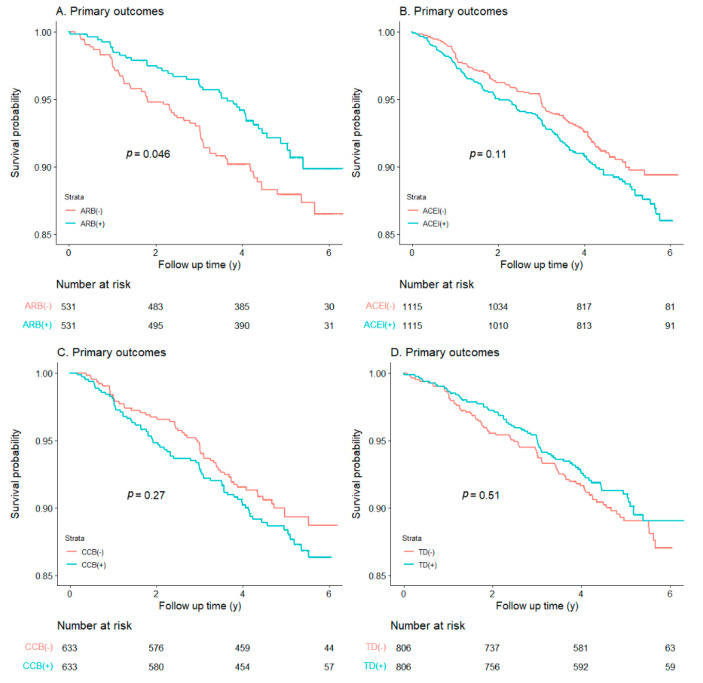
Kaplan−Meier survival analysis for primary outcomes within each antihypertensive therapy group. ((**A**): ARB; (**B**): ACEI; (**C**): CCB; (**D**): TD).

**Table 1 jcm-11-06486-t001:** Baseline characteristics of propensity score-matched patients with antihypertensive therapy and without.

**Characteristics**	**ARB**		** *p* ** **Value**	**ACEI**		** *p* ** **Value**
	**(−)**	**(+)**		**(−)**	**(+)**	
Number	531	531		1115	1115	
Age, mean (SD)	64.83 (6.95)	64.97 (6.74)	0.741	64.55 (6.74)	64.58 (6.63)	0.910
Female, no. (%)	273 (51.4)	254 (47.8)	0.269	496 (44.5)	458 (41.1)	0.113
Glycated hemoglobin, mean (SD)	8.11 (0.93)	8.13 (0.96)	0.694	8.23 (1.04)	8.24 (0.98)	0.890
BMI, mean (SD)	32.43 (5.27)	32.62 (5.52)	0.570	32.16 (5.51)	32.24 (5.30)	0.723
Smoke lifetime, no. (%)	243 (45.8)	245 (46.1)	0.951	547 (49.1)	546 (49.0)	1.000
Lipid, mean (SD)						
LDL	104.79 (33.72)	101.81 (33.15)	0.146	105.99 (34.18)	105.55 (33.29)	0.755
HDL	43.08 (12.34)	43.15 (12.35)	0.921	43.25 (12.03)	43.10 (11.77)	0.726
Triglyceride	183.73 (168.81)	186.82 (127.36)	0.737	187.71 (135.27)	186.28 (162.61)	0.822
BP, mean (SD)						
SBP	151.93 (10.57)	151.75 (11.10)	0.788	151.51 (10.53)	151.91 (10.74)	0.368
DBP	76.46 (8.25)	76.25 (8.18)	0.668	76.90 (7.74)	76.78 (8.10)	0.724
Race, no. (%)			0.322			0.896
White	293 (55.2)	310 (58.4)		683 (61.3)	687 (61.6)	
Non-White	238 (44.8)	221 (41.6)		432 (38.7)	428 (38.4)	
Education, no. (%)			0.700			0.261
Less than high school	84 (15.8)	73 (13.7)		176 (15.8)	192 (17.2)	
High school	140 (26.4)	151 (28.4)		317 (28.4)	298 (26.7)	
Some college	178 (33.5)	172 (32.4)		373 (33.5)	346 (31.0)	
College degree or higher	129 (24.3)	135 (25.4)		249 (22.3)	279 (25.0)	
Intensive glycemia therapy, no. (%)	255 (48.0)	267 (50.3)	0.500	562 (50.4)	561 (50.3)	1.000
Previous CVD, no. (%)	192 (32.3)	194 (36.5)	0.986	379 (34.0)	402 (36.1)	0.329
Previous heart failure, no. (%)	21 (4.0)	20 (3.8)	0.990	35 (3.1)	48 (4.3)	0.179
Medicines use, no. (%)						
Sulfonylureas	271 (51.2)	255 (48.0)	0.326	553 (49.6)	568 (50.9)	0.553
Biguanides	329 (62.2)	346 (65.2)	0.347	679 (60.9)	692 (62.1)	0.602
Meglitinide	14 (2.6)	11 (2.1)	0.679	28 (2.5)	33 (3.0)	0.604
Alpha-glucosidase inhibitors	0 (0.0)	5 (0.9)	0.074	1 (0.1)	4 (0.4)	0.371
Thiazolidinediones	114 (21.6)	133 (25.0)	0.203	258 (23.1)	237 (21.3)	0.308
Regular insulins	80 (15.1)	84 (15.8)	0.819	141 (12.6)	160 (14.3)	0.265
Statins	328 (62.0)	341 (64.2)	0.125	690 (61.9)	721 (64.7)	0.188
**Characteristics**	**CCB**		** *p* ** **Value**	**DT**		** *p* ** **Value**
	**(−)**	**(+)**		**(−)**	**(+)**	
Number	633	633		806	806	
Age, mean (SD)	65.76 (6.96)	65.38 (6.64)	0.321	64.86 (6.79)	64.72 (6.43)	0.680
Female, no. (%)	260 (41.1)	267 (42.2)	0.732	360 (44.7)	363 (45.0)	0.920
Glycated hemoglobin, mean (SD)	8.19 (0.99)	8.17 (0.92)	0.794	8.18 (1.00)	8.19 (0.93)	0.731
BMI, mean (SD)	32.23 (5.45)	32.48 (5.56)	0.404	32.57 (5.73)	32.76 (5.04)	0.467
Smoke lifetime, no. (%)	307 (48.5)	293 (46.3)	0.464	362 (44.9)	382 (47.4)	0.342
Lipid, mean (SD)						
LDL	100.32 (31.01)	102.59(31.87)	0.205	103.00 (33.12)	103.98 (34.63)	0.561
HDL	43.45 (12.41)	43.49 (11.66)	0.952	43.24 (12.46)	43.04 (11.29)	0.734
Triglyceride	182.68 (147.30)	178.13(159.05)	0.597	188.07 (155.43)	183.87 (129.08)	0.555
BP, mean (SD)						
SBP	153.00 (11.73)	152.40 (10.35)	0.335	152.18 (10.59)	152.12 (11.10)	0.916
DBP	75.73 (8.29)	75.39 (8.51)	0.472	76.55 (7.72)	76.61 (8.21)	0.876
Race, no. (%)			0.533			0.841
White	353 (55.8)	365 (57.7)		450 (55.8)	445 (55.2)	
Non-White	280 (44.2)	268 (42.3)		356 (44.2)	361 (44.8)	
Education, no. (%)			0.471			0.782
Less than high school	117 (18.5)	120 (19.0)		132 (16.4)	128 (15.9)	
High school	184 (29.1)	184 (29.1)		233 (28.9)	231 (28.7)	
Some college	191 (30.2)	193 (30.5)		263 (32.6)	252 (31.3)	
College degree or higher	141 (22.3)	136 (21.5)		178 (22.1)	195 (24.2)	
Intensive glycemia therapy, no. (%)	316 (49.9)	302 (47.7)	0.465	406 (50.4)	411 (51.0)	0.842
Previous CVD, no. (%)	266 (42.0)	259 (40.9)	0.732	303 (37.6)	290 (36.0)	0.535
Previous heart failure, no. (%)	29 (4.6)	28 (4.4)	1.000	16 (2.0)	22 (2.7)	0.412
Medicines use, no. (%)						
Sulfonylureas	321 (50.7)	322 (50.9)	1.000	399 (49.5)	404 (50.1)	0.842
Biguanides	410 (64.8)	403 (63.7)	0.725	533 (66.1)	535 (66.4)	0.958
Meglitinide	25 (3.9)	19 (3.0)	0.443	20 (2.5)	18 (2.2)	0.870
Alpha-glucosidase inhibitors	3 (0.5)	3 (0.5)	1.000	1 (0.1)	1 (0.1)	1.000
Thiazolidinediones	144 (22.7)	152 (24.0)	0.642	194 (24.1)	184 (22.8)	0.597
Regular Insulins	98 (15.5)	109 (17.2)	0.447	131 (16.3)	135 (16.7)	0.840
Statins	429 (67.8)	413 (65.2)	0.372	528 (65.5)	525 (65.1)	0.917

Continuous variables are expressed as mean (SD); categorical variables are expressed as frequency (percentage); ARB, angiotensin receptor blocker; ACEI, angiotensin-converting enzyme inhibitor; CCB, calcium channel blocker; TD, thiazide diuretics; BMI, Body Mass Index; LDL, low-density lipoprotein; HDL, high-density lipoprotein; BP, blood pressure; SBP, systolic blood pressure; DBP, dilated blood pressure; CVD, cardiovascular disease (including non-fatal myocardial infarction and stroke). Smoke lifetime means smoked more than 100 cigarettes during lifetime; ARB (+), patients treated with ARB; ARB (−), patients treated without ARB; ACEI (+), patients treated with ACEI; ACEI (−), patients treated without ACEI; CCB (+), patients treated with CCB; CCB (−), patients treated without CCB; TD (+), patients treated with TD; TD (−), patients treated without TD.

**Table 2 jcm-11-06486-t002:** Risks of primary outcomes for patients on different antihypertensive therapies compared to those without.

Antihypertensive Therapy	Model 1		Model 2		Model 3	
	HR (95% CI)	*p* Value	HR (95% CI)	*p* Value	HR (95% CI)	*p* Value
ARB (+)/ARB (−)	0.67 (0.45, 0.99)	0.047	0.66 (0.44, 0.98)	0.040	0.53 (0.34, 0.83)	0.006
ACEI (+)/ACEI (−)	1.24 (0.95, 1.61)	0.155	1.22 (0.94, 1.59)	0.126	1.19 (0.91, 1.54)	0.209
CCB (+)/CCB (−)	1.21 (0.86, 1.70)	0.268	1.27 (0.90, 1.78)	0.173	1.23 (0.87, 1.73)	0.245
TD (+)/TD (−)	0.89 (0.66, 1.23)	0.508	0.90 (0.66, 1.23)	0.503	0.88 (0.64, 1.21)	0.438

Model 1: unadjusted. Model 2: Adjusted for baseline age and gender. Model 3: Added race, education, Body Mass Index (BMI), smoking history, systolic and dilated blood pressure, HbA1c level, lipid level (low-density lipoprotein, high-density lipoprotein and triglyceride), therapies, previous heart failure, previous CVD, sulfonylureas, biguanides, meglitinide, alpha-glucosidase inhibitors, thiazolidinediones, regular insulins and statins to model 2. Abbreviations: ARB, angiotensin receptor blocker; ACEI, angiotensin-converting-enzyme inhibitor; CCB, calcium channel blocker; TD, thiazide diuretics; HR, hazard ratio; CI, confidence interval. ARB (+), patients treated with ARB; ARB (−), patients treated without ARB; ACEI (+), patients treated with ACEI; ACEI (−), patients treated without ACEI; CCB (+), patients treated with CCB; CCB (−), patients treated without CCB; TD (+), patients treated with TD; TD (−), patients treated without TD.

**Table 3 jcm-11-06486-t003:** Risk of primary outcomes for patients on ARB compared with those without ARB.

Outcomes	Model 1		Model 2		Model 3	
	HR (95% CI)	*p* Value	HR (95% CI)	*p* Value	HR (95% CI)	*p* Value
Cardiovascular mortality	0.31 (0.15, 0.66)	0.002	0.30 (0.14, 0.64)	0.002	0.32 (0.18, 0.90)	0.004
Non-fatal myocardial infarction	0.87 (0.53, 1.45)	0.605	0.87 (0.52, 1.44)	0.576	0.89 (0.53, 1.52)	0.692
Non-fatal stroke	1.01 (0.38, 2.71)	0.977	1.01 (0.38, 2.72)	0.977	1.04 (0.38, 2.84)	0.933

Model 1: Unadjusted. Model 2: Adjusted for baseline age and gender. Model 3: Added race, education, Body Mass Index (BMI), smoking history, systolic and dilated blood pressure, HbA1c level, lipid level (low-density lipoprotein, high-density lipoprotein and triglyceride), therapies, previous heart failure, previous CVD, sulfonylureas, biguanides, meglitinide, alpha-glucosidase inhibitors, thiazolidinediones, regular insulins and statins to model 2. Abbreviations: ARB, angiotensin receptor blocker; HR, hazard ratio; CI, confidence interval.

## Data Availability

No data are available. The ethical statement and the informed consent do not allow for free data availability.

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
