# Peer review of "The Impact of Different Antihypertensive Drugs on Cardiovascular Risk in Isolated Systolic Hypertension with Type 2 Diabetes Patients"

_jcm, 2022, doi:10.3390/jcm11216486_

Round 1
Reviewer 1 Report
Using data of the "Action to Control Risk in Diabetes" (ACCORD) trial the relelationship between antihypertensive drugs and cardiovascular risk in patients with isolated systolic hypertension (ISH) and type 2 diabetes mellitus (T2DM) has been investigated. By evaluating the data of interest propensity score matching was used and Kaplan Meier survival analyses were performed.
The primary outcome (PO) was a combined endpoint defined as the first occurrence of cardiovascular death (CM) or non-fatal myocardial infarction (NMI) or non-fatal stroke (NST). Secondary endpoints were cardiovascular mortality (CM), non-fatal myocardial infaction (NMI) and non-fatal stroke (NST).
Major comments:
Allthough the data may be of clinical and pathophysiological interest there are some major issues to be clarified before a final judgement can be provided:
- Design and Methods of the ACCORD trial were already published about 15 years ago (2007). Therefore the time period of patient`s recruitment and medical treatment is essential to know, as meanwhile other treatment options are available like SGLT-2 inhibitors.
- Moreover, selective beta-blockers have not been taken into consideration, which especially may be of interest in patients with concomitant cardiac diseases (like patients with a history of acute coronary syndrome, patients with chronic heart failure, patients with atrial fibrillation etc.)
- Against this background (a) the time period (date) of the patient`s recruitment and (b) the percentage of included patients with cardiovascular disease would be of interest and should be included into the study.
- On which basis has the diagnosis of "isolated systolic hypertension" been confirmed? Also please provide data on the incidence and prevalence of ISH in the general population as compared to the actually evaluated population of DM type 2 patients in the ACCORD registry. Such data are of importance for the reader to deliver some idea on the actual clinical relevance of the presented data.
- A detailed CONSORT-flow diagram of the patient`s recruitment must definetively be presented within the manuscript.
Detailled comments:
Introduction:
- Line 34: The term "prevalence rate" is not correct and should be simply transferred to the term "prevalence".
- Line 34: include a number reflecting the prevalence of ISH as reported in literature
- Lines 36-37: the term "diabetes complications" should be specified
- The authors state that "it is particularly important to control blood pressure in T2DM patients with ISH". This statement may be misleading, as patients with systolic + diastoloc hypertension are also at increased risk, especially if they additionally suffer from diabetes.
- Lines 47-48: please provide literature/references on the discussed hypertensive mechanisme in ISH, especially in patients with diabetes.
Methods:
Interventions: The authors restricted the treatment options to ARB, ACEI, CCB and TD, but did not consider beta-blockers and aldostenone antagonists. This should be commented/justified within the Methods section.
Primary endpoint (PE): the PE includes competing single endpoints, which especially is a problem, if fatal endpoints have not been separately evaluated. In this case a "fatal event"even shortly occurring after a first "non fatal "event would not have been considered. This problem, however, has correctly been solved in this study by a separate evaluation of the single endpoints, showing a significant reduction of CV-mortality but not of NMI of NST. Unfortunatelly, the endpoint "total mortality" ("death of any cause") has not been presented, which should be included into the calculations and data presentations.
Results:
- Table 1: this table is confusing and must be rearranged. Especially provide a clear subdivision between the groups of ARB, ACEI, CCB, TD.
- Table 1: Major clinical characteristics are lacking and must be included (e.g. history of major clinical events like acute coronary syndrome, heart failure, stroke). Apart from the anti-hypertensive drugs under investigation there is no information on other medications like statins, antidiabetics etc. . all of which do have prognostic relevance! I strongly recommed to include such data.
Discussion:
- lines 205-217: By reporting and discussing the differential pathophysiological and clinical effects of ACEI and ARB it would be advisable to also refer to actual guidelines. This especially is important in case of controversial outcomes that need to be discussed.
lines 242 - 244: This paragraph ( "For another, TD can cause metabolic disorders....") appears to be vague and somewhat "superficial", and therefore should be substantiated. Which "metabolic disorders" are referred to?
line 246: The sentence "first, we minimized confusion...." --> please replace by "...potential bias has been reduced by the application of propensity score matching..."
Author Response
Thank you for your comments concerning our manuscript. These comments are valuable and very helpful. We have read through the comments carefully and have made the corresponding corrections:
A1:Thank you for your suggestions and opinions. We are very sorry that we did not clearly describe the recruitment time and the main medical treatment in the materials and methods section. We have made a supplementary explanation again, as follow:
“ACCORD main trial recruitment started in February 2003. The final visit for the last randomized participant was ended on June 30, 2009, with each participant receiving 4 to 8 years of treatment and follow-up (approximate mean, 5.6 years). All participants were randomly assigned to groups standard glycemia control (HbA1c target of 7%-7.9%) or intensive glycemia control (HbA1c target of ≤ 6.0%). Hypoglycemic medical therapy mainly includes sulfonylureas, biguanides, meglitinide, alpha-glucosidase inhibitors, thiazolidinediones, regular insulins and so on.”
Unfortunately, SGLT-2 inhibitors were not included in the ACCORD study.
A2: We strongly agree with your opinion that selective beta-blockers have good therapeutic effect for patients with concomitant cardiac diseases (like patients with a history of acute coronary syndrome, patients with chronic heart failure, patients with atrial fibrillation etc). But we didn't take it into account for two reasons:
- Our study population were T2DM with ISH patients. According to the 2017 ACC/AHA/AAPA/ABC/ACPM/AGS/APhA/ASH/ASPC/NMA/PCNA Guideline for the Prevention, Detection, Evaluation, And Management of High Blood Pressure in Adults (DOI: 10.1161/HYP.0000000000000065), only recommend ARB, ACEI, CCB and TD as first-line antihypertensive medication for T2DM patients.
- Due to the limitation of ACCORD database, the specific types of β-blockers (β1 or β2) were not distinguished and were uniformly expressed as β-blockers, which made subsequent detailed analysis difficult.
A3: Thank you for your suggestion. (a) We have added the time period (date) of the patient's recruitmen in the Materials and Methods section (This is explained in A1). (b) the percentage of included patients with cardiovascular disease was supplemented in baseline Table 1. The specific contents are as follows:
“Previous cardiovascular disease (include non-fatal myocardial infarction and stroke): with ARB:192 (32.3%), without ARB:194 (36.5%) p = 0.986; with ACEI:379 (34.0%), without ACEI:402 (36.1%) p = 0.329; with CCB:266 (42.0%), without CCB:259 (40.9%) p = 0.732; with TD:303 (37.6%), without TD:290 (36.0%) p = 0.535”
“Previous heart failure: with ARB:21 (4.0%), without ARB:20 (3.8%) p = 0.990; with ACEI:35 (3.1%), without ACEI:48 (4.3%) p = 0.179; with CCB:29 (4.6%), without CCB:28 (4.4%) p = 1.000; with TD:16 (2.0%), without TD:22 (2.7%) p = 0.412”
A4:According to 2017 ACC/AHA/AAPA/ABC/ACPM/AGS/APhA/ASH/ASPC/NMA/PCNA guideline for the prevention, detection, evaluation, and management of high blood pressure in adults (DOI: 10.1016/j.jacc.2017.11.006) and 2018 ESC/ESH Guidelines for the management of arterial hypertension (DOI: 10.1093/eurheartj/ehy339), the isolated systolic hypertension was defined as systolic blood pressure ≥ 140 mmHg and diastolic blood pressure < 90 mmHg. We provide a supplementary explanation in the Materials and Methods section.
The prevalence of ISH in the general population ranges from 1.9% to 4.3%. (DOI: 10.1007/s11906-016-0686-x; DOI: 10.1038/sj.jhh.1002119; DOI: 10.1111/j.1463-1326.2005.00523.x), and we have made a supplementary explanation in introduction section. A total of 10251 participants with T2DM were enrolled in the ACCORD trial(DOI:10.1016/j.amjcard.2007.03.003), the number of people who met the criteria of isolated systolic hypertension were 2547, and the prevalence was about 24.84%. We describe this in the Materials and Methods section.
A5: Thanks for your suggestion, we have made a detailed CONSORT-flow diagram of the patient's recruitment, which has been re-uploaded as Supplementary Figure 1 in the end of the manuscript.
Detailled comments:
Introduction:
- Thank you for reminding us. We have changed "prevalence rate" to "prevalence".
- We added data on the prevalence of ISH in introduction as follows:
“The prevalence of ISH in the general population ranges from 1.9% to 4.3%. (DOI: 10.1007/s11906-016-0686-x; DOI: 10.1038/sj.jhh.1002119; DOI: 10.1111/j.1463-1326.2005.00523.x). In a cross-sectional study of Chinese patients with type 2 diabetes, the prevalence of ISH was found to be 7.6%, more than twofold higher than that of non-diabetic patients (3.4%)”(DOI: 10.1111/j.1463-1326.2005.00523.x)
- According to the references (DOI:10.1136/bmj.321.7258.412), complications related to diabetes include myocardial infarction, sudden death, angina, stroke, renal failure, lower extremity amputation or death from peripheral vascular disease, death from hyperglycaemia or hypoglycaemia, heart failure, vitreous haemorrhage, retinal photocoagulation, and cataract extraction. We have detailed this in section Introduction.
- Thank you for your suggestion to our less rigorous expression. We have modified it as follows:
“Decreasing of systolic blood pressure can significantly improve the prognosis of diabetes and a decrease of 10 mmHg leads to 12% reductions of diabetes complications (myocardial infarction, sudden death, angina, stroke, renal failure, lower extremity amputation or death from peripheral vascular disease, death from hyperglycaemia or hypoglycaemia, heart failure, vitreous haemorrhage, retinal photocoagulation, and cataract extraction). Therefore, it is important for improving prognosis to control blood pressure in T2DM patients.”
- We mentioned that on Lines 47-48: T2DM with ISH patients have greater pulse pressure, higher degree of arterial blood wall stiffness, more serious degree of vascular aging, and greater cardiovascular risk.
According to the original text in the reference (DOI: 10.1016/j.ijcard.2004.03.059), we can support our opinion, as follows:
“Large-scale epidemiological surveys have confirmed that systolic blood pressure (SBP) was more strongly associated with cardiovascular risk than diastolic blood pressure (DBP)”
“It is usually found in elderly who have degenerated and stiffened vasculatures. With aging, arteries lose their elasticity with reduced compliance to stroke volume leading to increased SBP. The lowest arterial capacitive compliance has been demonstrated in patients with ISH”
“Moreover, among the diabetic subjects, duration of diabetes was an independent predictor for ISH suggesting that prolonged hyperglycaemia might play an important role in the pathogenesis of ISH. In accord to this hypothesis, a novel non-enzymatic breaker of advanced glycation end product has been shown to selectively improve arterial compliance and reduce pulse pressure in non-diabetic elderly subjects with vascular stiffening”
“Our findings suggest that apart from age, neurohormonal abnormalities and metabolic derangement might all contribute to the early degeneration and stiffening of blood vessel walls.”
Methods:
- Thank you for your advice. We have explained the reasons why only ARB, ACEI, CCB and TD are selected (in A2), and have made supplementary explanations in the section of Materials and Methods.
- Thank you for your suggestion that we take "total mortality" into account again. In Supplementary Table 1, we added the effect of four antihypertensive drugs on the total mortality of T2DM with ISH patients, but unfortunately, they did not reduce the total mortality (Model 3: ARB: p = 0.125; ACEI: p = 0.137; CCB: p = 0.759; TD: p = 0.876). We uploaded supplementary Table 1 at the end of the manuscript.
Results:
- Thanks for your reminding. We have modified Table 1 again to provide a clear subdivision between the groups of ARB, ACEI, CCB and TD.
- Thanks for your suggestion. We included previous cardiovascular disease (include non-fatal myocardial infarction and stroke), previous heart failure, antidiabetics (sulfonylureas, biguanides, meglitinide, alpha-glucosidase inhibitors, thiazolidinediones and regular insulins) and statins in Table1 for statistical analysis again. They were evenly matched in the antihypertensive groups (with ARB, ACEI, CCB or TD) and non-antihypertensive groups (without ARB, ACEI, CCB or TD).
The results did not change substantially: the cumulative incidence rate of primary outcomes (PO) in the use ARB group were significantly lower than those without (hazard ratio (HR) 0.53; 95% confidence interval (CI) 0.34–0.83; P = 0.006). But for ACEI, CCB and TD, were negligible (ACEI: P = 0.209; CCB: P = 0.245; TD: P = 0.438). ARB decreased Cardiovascular mortality (CM) in PO rather than non‐fatal myocardial infarction (NMI) and non-fatal stroke (NST) (CM: HR 0.32; 95%CI 0.18-0.90; P = 0.004; NMI: P = 0.692; NST: P = 0.933).
Discussion:
- Thank you very much for your suggestion. By reviewing the recent ACC/AHA/ESC guidelines on antihypertensive drugs, we found that there is no difference in the recommendation level of these two drugs (ACEI and ARB) at the guideline level. They are both first-line antihypertensive drugs for diabetic patients, especially for patients with chronic kidney disease and proteinuria. However, ACEI has the side effect of irritating cough, ARB is recommended for such patients. (DOI: 10.1161/CIR.0000000000000597; DOI: 10.1016/j.rec.2018.12.004; DOI: 10.1111/jch.12638; DOI: 10.1016/j.tcm.2019.05.003) Unfortunately, no large RCTS or meta-analyses have explored the differences between the two. Therefore, further studies are needed to confirm the reliability of the conclusions drawn in our study.
- Thank you for your suggestions and comments. We explained the definition of "metabolic disorders" in more detail. TD can cause electrolyte disturbances, leading to hypokalemia, hyponatremia, and hypomagnesemia (DOI: 10.1016/j.amjmed.2021.04.007). TD can also affect fasting blood glucose and lead to deterioration of glucose tolerance (DOI: 10.1007/s11892-018-0976-6), aggravate hepatic steatosis, such as liver triglyceride content, leading to visceral fat accumulation, and reduce insulin sensitivity (DOI:10.1186/1758-5996-5-35; DOI: 10.1161/HYPERTENSIONAHA.108.119404). We also explain this in the manuscript.
- Thank you for your suggestion, and we have corrected our inappropriate expression.

Reviewer 2 Report
Well performed study on an important topic. Novelty is limited by use of data from study published in 2007 but overall worthwhile given importance of data and results.
Author Response
Thank you for your recognition of our manuscript.
Reviewer 3 Report
I received for review an article prepared by Ming Gao et al. entitled "The impact of different antihypertensive drugs on cardiovascular risk in isolated systolic hypertension with type 2 diabetes patients", which is being processed for publication in the Journal of Clinical Medicine (IF=4,964). The work concerns the extremely important topic of cardiovascular risk in patients with type 2 diabetes. Nowadays, cardiovascular diseases are the main cause of morbidity and mortality in the world, and type 2 diabetes is one of the most important cardiovascular risk factors. The presented original paper has a cognitive value and should be considered for publication in the future, but in my opinion significant corrections are necessary, thanks to which the quality of the manuscript may be improved.
1) The introduction should be extended with additional information. In the first paragraph, the authors write about pulse pressure, which is the basic parameter in the assessment of vessel stiffness. It is worth mentioning the issue of vascular stiffness, the assessment of which is widely discussed, inter alia, in the population of patients with metabolic syndrome, an important element of which is carbohydrate metabolism disorders. It is worth mentioning that diabetes is a significant risk factor for the development of atherosclerosis. In the course of atherosclerosis, coronary artery disease, cerebrovascular disease and peripheral arterial disease develop. Percutaneous balloon angioplasty procedures with possible stent implantation play an important role in the treatment of these diseases, but diabetes is a significant risk factor for restenosis, which worsens the prognosis and may lead to the need for reintervention. Atherosclerosis in patients with diabetes is slightly different than in non-diabetic patients. In peripheral arterial disease, lesions are usually multi-level and often more severe in the arteries below the knee. (see e.g. the following items: 10.3390/ijerph191610368; 10.3390/ijerph182211970; 10.3390/ijerph17249339; 10.4239/wjd.v6.i7.961; 10.2217/fca-2018-0045; 10.1007/s11886-019-1107-y; 10.1016/j.phrs.2016.09.040).
2) It should be not “angiotensin type 2 antagonists”, but “angiotensin receptor blockers” or more formally “angiotensin II receptor type 1 (AT1) antagonists”.
3) It should be not “pulse pressure difference” but only “pulse pressure” because pulse pressure is defined as the difference between systolic blood pressure and diastolic blood pressure. (line 48)
4) The purpose of the study should be precisely defined at the end of the introduction.
5) The methodology of the statistical analysis should describe how the consistency of the distribution of quantitative variables with the normal distribution was tested.
6) Table 1 is a bit unreadable, it should be fine-tuned in terms of editing and visual.
7) For quantitative variables whose distribution is inconsistent with the normal distribution, it is better to use the median and the interquartile range, while the arithmetic mean and standard deviation are better parameters for variables whose distribution is consistent with the normal distribution.
8) In my opinion, it would be better “antihypertensive role” than “hypotensive role” (line 206).
9) The text should be revised in terms of editing, style and language. In some places in the text, some words in the middle of the sentence are written with capital letter, in my opinion unjustified.
10) In some places, the text lacks reference to literature, e.g. from lines 239 to 245.
11) The reference list should be prepared in accordance with the MDPI rules.
Author Response
Thank you for your comments concerning our manuscript. These comments are valuable and very helpful. We have read through the comments carefully and have made the corresponding corrections:
- Thank you for your suggestions to us. We have added the following contents and cited the corresponding references as follows:
“The high cardiovascular risk of people with diabetes is reflected in the fact that diabetes can cause atherosclerosis, leading to coronary artery disease, cerebrovascular disease, peripheral artery disease and other cardiovascular diseases.” (DOI: 10.1016/j.phrs.2016.09.040; DOI: 10.2217/fca-2018-0045; DOI: 10.1007/s11886-019-1107-y)
“The cause of elevated pulse pressure in T2DM patients is related to increased vascular stiffness caused by impaired carbohydrate metabolism. (DOI:10.3390/ijerph191610368) This may be the reason why people with type 2 diabetes are more likely to develop ISH.”
- Thank you for your suggestions on our inappropriate expression. We have revised "angiotensin type 2 antagonists" to "angiotensin receptor blockers (ARB)".
- Thank you for your suggestion. We have corrected "pulse pressure difference" to "pulse pressure".
- Thank you for your reminding. At the end of the INTRODUCTION, we put forward our purpose objectives as follows:
“We decided to use the data from ACCORD trial for a post-hoc analysis to investigate the effects of four antihypertensive drugs (ARB, ACEI, CCB or TD) on the cardiovascular prognosis of ISH with T2DM patients, so as to provide scientific basis for clinical rational drug use.”
- The literature What is so normal about the normal distribution? (DOI: 10.1136/ebmh.13.4.100) points out that approximations to normal distributions occur in many situations in medicine. Examples include height, weight and blood pressure. The central limit theorem states that when a large number of samples (of reasonable size) is drawn from a population, the distribution of the sample means will be approximately normal, whatever the distribution of the data in the population. If the distribution is markedly different from normal, a sample size of at least 30 would be necessary. In our study, a sample size of more than 30 can be considered as conforming to a normal distribution. We also describe this in our Methods section. Thanks for reminding.
- Thanks for your reminding. We have again revised Table 1 to ensure brevity and readability.
- Thank you for your advice. As mentioned in point 5, the quantitative variables in our study approximately fit a normal distribution, so we used arithmetic means and standard deviations to describe our data.
- Thanks for your reminding, we have modified "hypotensive role" to "antihypertensive role".
- Thank you for your suggestion. We have modified the capital letter in the middle of the sentence.
“In 21, 83, 136, 137, 162, 213 lines, Primary changed to primary”
“In 48 lines, Angiotensin changed to angiotensin”
“In 50 lines, Calcium changed to calcium”
“In 75 lines, Cardiovascular changed to cardiovascular”
“In 76 lines, Non changed to non”
- Thank you for your suggestions and comments. We explained the definition of "metabolic disorders" in more detail. TD can cause electrolyte disturbances, leading to hypokalemia, hyponatremia, and hypomagnesemia (DOI: 10.1016/j.amjmed.2021.04.007). TD can also affect fasting blood glucose and lead to deterioration of glucose tolerance (DOI: 10.1007/s11892-018-0976-6), aggravate hepatic steatosis, such as liver triglyceride content, leading to visceral fat accumulation, and reduce insulin sensitivity (DOI:10.1186/1758-5996-5-35; DOI: 10.1161/HYPERTENSIONAHA.108.119404). We also explain this in the manuscript.
- Thank you for your suggestions. We have revised the format of the reference in accordance with MDPI rules to the format of "Chicago 17th Footnote", revised the chart and added the CONSORT-flow diagram as supplementary Figure 1, which was uploaded at the end of the manuscript.

Round 2
Reviewer 1 Report
Dear authors,
as a results of detailled revisions the manuscript has markedly been improved. As a minor comment please note that you should define the "primary outcome" wthin the abstract (check line 21).
Author Response
Thank you for your advice. In line 21, we modified it as follows:
“The cumulative incidence rate of primary outcomes (PO, include cardiovascular mortality, non‐fatal myocardial infarction and non‐fatal stroke)”

Reviewer 3 Report
The paper has been improved. I recommend it for publication in its current form.
Author Response
Thank you again for your recognition of our manuscript.